

# First observations of continuum emission in dayside aurora

Noora Partamies[1], Rowan Dayton-Oxland[2], Katie Herlingshaw[1], Ilkka Virtanen[3],
Bea Gallardo-Lacourt[4,5], Mikko Syrjäsuo[1], Fred Sigernes[1], Takanori Nishiyama[6], Toshi Nishimura[7],
Mathieu Barthelemy[8], Anasuya Aruliah[9], Daniel Whiter[2], Lena Mielke[1], Maxime Grandin[10],
Eero Karvinen[11], Marjan Spijkers[12], and Vincent E. Ledvina[13]

[1]Depertment of Arctic Geophysics, The University Centre in Svalbard, Longyearbyen, Norway
[2]Department of Physics and Astronomy, University of Southampton, Southampton, UK
[3]Space Physics and Astronomy, University of Oulu, Oulu, Finland
[4]NASA Goddard Space Flight Center, Greenbelt, 20771, MD, USA
[5]Department of Physics, The Catholic University of America, N.E. Washington, 20064, DC, USA
[6]National Institute of Polar Research, Japan
[7]Boston University College of Engineering, 725 Commonwealth Avenue, Boston, MA 02215, USA
[8]University of Grenoble Alpes, CNRS, IPAG, 38000 Grenoble, France
[9]Department of Physics and Astronomy, University College London, London, UK
[10]Finnish Meteorological Institute, Helsinki, Finland
[11]URSA, Skywarden service, Finland
[12]Noorderlicht en meer, Zomerdijkstraat 23, 8043 HW Zwolle, Netherlands
[13]University of Alaska Fairbanks, Fairbanks, Alaska, USA

**Correspondence:** Noora Partamies (noora.partamies@unis.no)

**Abstract.** We report the first observations of continuum emission at the poleward boundary of the dayside auroral oval. Spectral measurements of high-latitude continuum emissions resemble those of STEVE, with light characterised by colours such as white, pale pink or mauve. The emission enhancement spans the entire visible wavelength range. However, unlike STEVE, the high-latitude dayside continuum emission events tightly follow the auroral particle precipitation often forming field-aligned

rays and other dynamic shapes. Some dayside emissions appeared as wide arcs or cloud-like structures within the red-emission dominated dayside aurora. Our spectral measurements further suggest that the broad band continuum emission may extend into the near-infrared regime. Similar to the STEVE emission, low-Earth orbit measurements of plasma flow in the region of continuum emission show a strong horizontal cross-track velocity shear. Ground-based radar and optical observations provide evidence of both plasma and neutral heating, as well as upwelling, in connection to the continuum emissions. We conclude that

the interplay of different heating mechanisms may be an important factor in generating high-latitude continuum emissions.

## 1 Introduction

Continuum emissions in the night sky have been observed for decades. One of the earliest reports by Meinel (1953) describes continuum emissions observed in wavelengths between 390 nm and 480 nm. In nightglow studies, in particular, continuum emission has been recognized as a well-known emission type (Noll et al., 2024). In recent years, continuum emissions have

also gained interest in the auroral community following the discovery of a new optical phenomenon known as Strong Thermal



Emission Velocity Enhancement or STEVE (MacDonald et al., 2018). STEVE appears as a mauve-white arc just equator-ward of the auroral oval in the sub-auroral ionosphere. STEVE is narrow (∼0.5° latitudinal extent) and spans several hours in magnetic local time (MLT) (Gallardo-Lacourt et al., 2018a). Satellite observations have identified STEVE as the optical manifestation of extreme sub-auroral ion drifts (Archer et al., 2019a). Initially, STEVE was identified to occur during active
geomagnetic conditions, consistent with storm-time conditions. However, more recent observations have reported STEVE dur-ing quiet geomagnetic conditions, raising questions about the generation mechanism of these intense sub-auroral ion drifts and the role of the magnetosphere in the process (Gallardo-Lacourt et al., 2018a; Martinis et al., 2022; Gallardo-Lacourt et al., 2024; Nishimura et al., 2024). Interestingly, particle detectors onboard low-Earth orbit satellites have determined that STEVE is not related to particle precipitation, suggesting that STEVE emissions could be generated locally in the ionosphere
(Gallardo-Lacourt et al., 2018b; Nishimura et al., 2019). Consistently, field-aligned current (FAC) measurements by satellites have confirmed the presence of a downward FAC collocated with STEVE observations (Archer et al., 2019a; Nishimura et al., 2019). A more recent report on STEVE-like pale emissions located poleward of the green morning sector aurora (Nanjo et al., 2024) suggests that strong gradients at the boundary of Region-1 and Region-2 currents are critical for generating strong ion flows and heating.

These significant discrepancies with traditional auroral processes prompted further research into STEVE's spectral charac-
teristics. Gillies et al. (2019) determined STEVE's spectrum for the first time using a spectrograph from the Transition Region Explorer (TREx) array capable of analyzing emissions between 400 nm and 800 nm. These analyses revealed a continuum emission with elevated intensities at all measured wavelengths, solidifying the understanding that STEVE is not produced by particle precipitation but potentially by local ionospheric processes. Further analysis of STEVE's spectrum and optical properties supports these findings (Mende and Turner, 2019; Gillies et al., 2023).

One of the most intriguing questions in STEVE research is understanding the processes involved in its generation and unusual
spectrum. Harding et al. (2020) used a simple photochemical model to demonstrate that the fast sub-auroral ion drifts (SAIDs) measured alongside STEVE observations could excite nitrogen molecules into vibrationally excited states. This then results in nitrogen oxide that combines with ambient oxygen to produce $NO_2$ and a spectrally continuous emission. Additionally,
feedback-unstable magnetosphere-ionosphere interactions have been evaluated as possible generation mechanisms for STEVE (Mishin and Streltsov, 2019), challenging the photochemical mechanism proposed by Harding et al. (2020). Currently, there is no consensus on how these continuous emissions are created, and measurements are scarce. Future missions, such as NASA's planned Geospace Dynamics Constellation (GDC) and Dynamics missions, could help elucidate the physical mechanisms at play in STEVE from the magnetosphere–ionosphere coupling perspective.

Recent studies have revealed the presence of auroral structures with continuous emissions within the auroral oval, differ-ent from STEVE. Using the TREx colour imagers (RGB) and spectrograph array, researchers have identified these emissions, which would otherwise be overlooked or mistaken for single-wavelength emissions in panchromatic cameras. In the absence of full spectral measurements, continuum emissions could easily be misinterpreted as enhancements of individual emission lines, or a combination of individual emission lines and certain conditions of background sky illumination on the same line-of-sight.
Interestingly, these new observations of continuum emissions are closely associated with auroral dynamics and precipitation.





Events captured by the TREx RGB cameras, when compared to spectrograph data, not only confirmed the presence of continuum emissions but also uncovered embedded auroral emissions within the same region (Spanswick et al., 2024). While more events are needed to understand the frequency, spatial distribution and other characteristics of these continuum emissions, they appear to be not uncommon within the auroral oval but rather overlooked due to the instrumentation designed to measure
specific wavelengths.

This study reports the first observations of continuum emissions detected around the polar cap boundary over Svalbard (78°N) in arctic Norway. The continuum emission events reported in this paper occurred within the dayside aurora, but we also show events in the afternoon and nightside aurora, which are likely to be continuum emissions. Our observations align with those of Spanswick et al. (2024) in that they follow the particle precipitation and auroral dynamics. They may also be
influenced by strong gradients at the poleward boundary of the auroral oval, as reported by Nanjo et al. (2024), and could even associated with strong plasma flow shears, similar to STEVE.

In the following sections, we describe our observational setup (Section 2) and our four sample events (Section 3). The sample events were chosen based on the data coverage and their MLT occurrence to demonstrate the variety of different ionospheric conditions in which continuum emissions may appear. Sections 4 & 5 are dedicated to the discussion of the results
and conclusions, respectively.

## 2  Data and instrumentation

As the continuum emission is enhanced across the entire visible wavelength range, the best way to observe it is with instrumentation covering the whole spectral range. We use a full-colour all-sky camera (ASC) and an imaging spectrograph. In full-colour images, the color of the continuum emission does not correspond to any known individual emission line. An ex-
ample of this can be seen in Figure 1a as an arc-like region of pale pink colour. To confirm the correct identification of the continuum emission, spectral measurements are necessary. An example of the visible range spectra, collected along the local magnetic meridian and corresponding to the centre column of the all-sky image (white rectangle), is shown in Figure 1b. The spectra around the scan angle of 120 (indicated by the black rectangle) show enhancement at auroral emission wavelengths of 428, 558 and 630 nm, as well as across all the displayed wavelengths from 400 to 700 nm.
Kjell Henriksen Observatory (KHO, 78°N) hosts a full-colour mirrorless camera (Sony $\alpha$7S, for details, see e.g. Dreyer et al. (2021)) as well as a the Meridian Imaging Svalbard Spectrograph (MISS) described in a recent KHO science review by Herlingshaw et al. (2024). The camera has been in operation since 2015, but has been preceded by another full-colour DSLR since 2008. The current imaging cadence is about 12 seconds with a nighttime exposure of 4 seconds. The full pixel resolution of the images is 2832×2832.
The MISS spectrograph, installed in late 2019, provides a spectral image along the magnetic meridian every minute. The optics incorporate a 0.1 mm slit, which provides a roughly one degree wide field-of-view along the meridian. The spectral resolution capability is 1.5 nm. However, due to the internal optics being out of focus, the spectral resolution of the data used in this study was deteriorated to about 5 nm, which still works well for the purpose of this study. The data are wavelength



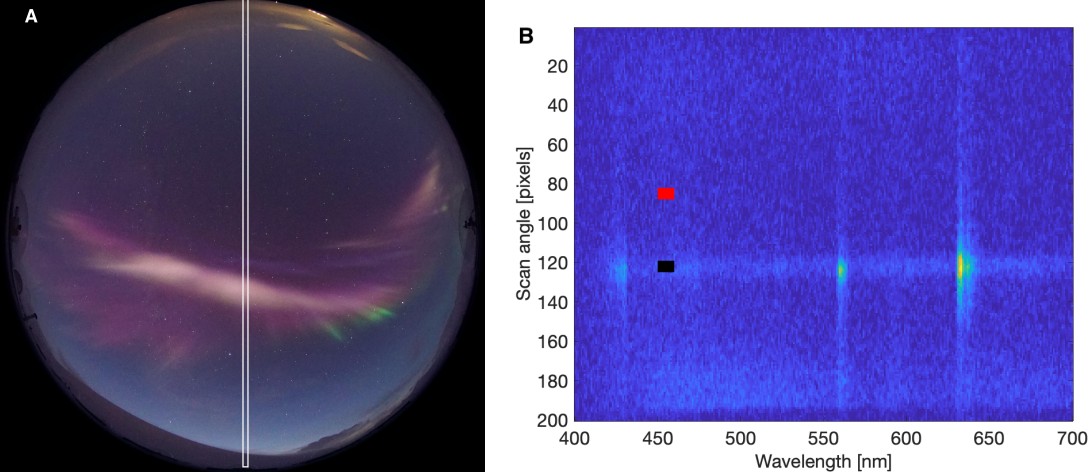

**Figure 1.** A: A full-colour all-sky image from Kjell Henriksen Observatory (KHO), Svalbard, exemplifying the pale pink continuum emission as an arc-like structure in the middle of red-dominated auroral emission. The image is take on 3 January 2020 at 08:48:56 UT. North is to the top and east is to the left of the image. The white rectangle marks the location of the spectrometer field-of-view in the image. The Sun was located at an elevation angle of about -13° relative to the horizon with an azimuth of 149° (SSE). The ionosphere above about 160 km altitude was sunlit. B: Spectra measured by Meridian Imaging Svalbard Spectrograph (MISS) at 08:49 UT. Spectra across the visible wavelength range at 400–700 nm are collected along the magnetic meridian. The continuum emission in the image corresponds to the spectral enhancement at the scan angles of about 120, marked by the black square. An empty sky reference scan angle is indicated by the red square at the scan angle of about 90.

calibrated using the three strongest auroral emission lines at 428, 558 and 630 nm as references, with a second-order polynomial
fitted to the wavelength regions in between. The CCD background has been subtracted based on pixel values outside the slit region (cropped out of the displayed data), and no absolute intensity calibration is done for these data.

Additional spectral information has been gathered from the Near-InfraRed Aurora and airglow Camera (NIRAC) that measures molecular nitrogen ion emission of $N_2^+$ Meinel (0,0) band at 1 $\mu$m (Nishiyama et al., 2024). This emission is primarily prompted by electrons penetrating down to about 100 km altitude, which corresponds to precipitating electron energies up to
10 keV. The emission can also be generated above 140 km by charge exchange between molecular nitrogen ($N_2$) and atomic oxygen ($O^+$). NIRAC's field-of-view is 84×68° and the nominal temporal resolution is 20 seconds.

High-resolution spectra have been obtained by the High-Throughput Imaging Echelle Spectrograph (HiTIES, Chakrabarti et al. 2001), located at the KHO. The instrument includes an Echelle spectrograph grating, an electron-multiplying charged
coupled device (EMCCD) detector, and a mosaic filter which is used to select multiple overlapping spectral orders, enabling observation of multiple non-contiguous wavelength bands at high resolution (< 0.1 nm). The mosaic filter selects the H-alpha band at 649–663 nm used for observing proton precipitation, $O^+$ green line / OH(8,3) band at 728–743 nm, and OH(5,1) band at 790–807 nm. HiTIES is directed at the magnetic zenith with an 8° by 0.05° field-of-view. The imaging cadence is 2 Hz,




giving a temporal resolution of 0.5 seconds. The spectra in this study have been spatially and temporally integrated to improve the signal-to-noise ratio. The background has been subtracted based on pixel values outside of the filter mosaic, and the data are not calibrated for absolute intensity. Dark frames are taken every 20 minutes and flat fields are taken at the beginning of the observing season.

The EISCAT Svalbard Radar (ESR) (Wannberg et al., 1997) is located about 1 km north of KHO, and provides profiles of ionospheric plasma parameters within the zenith region of the Sony camera field-of-view. The data consist of height profiles of electron density, electron and ion temperature and ion velocity as a standard set of parameters along the radar beam with a temporal resolution of one minute. The radar experiment that was run during one of our events was `ipy`, which has a height resolution of about 4–5 km in the ionospheric E region. In this study, we only use data from the non-steerable parabolic antenna (42 m in diameter), which measures plasma profiles along the geomagnetic field direction.

The Fabry-Perot Interferometer (FPI) at KHO can determine the neutral winds and temperatures in the thermosphere at the heights of red auroral and airglow emission (630.0 nm), which typically has a peak emission at around 240–250 km (Aruliah and Griffin, 2001). The winds and temperatures are based on Doppler shift and Doppler broadening of the observed emission. The FPI has well-separated look directions towards zenith, north, east, south and west at an elevation angle of 30°. The wind and temperature estimates have a temporal resolution of 8 min. The line-of-sight wind measurements can be converted to horizontal estimates by assuming that there is zero vertical wind. In this study, we only use the zenith line-of-sight location to estimate the neutral upwelling.

The Defense Meteorological Satellite Program (DMSP) F17 satellite flew over one of our continuum emission events. In our analysis, we use the horizontal cross-track plasma flow measurements from F17 (Cornelius and Mazzella, 1994). These data are archived in the Coupling Energetics and Dynamics of Atmospheric Regions (CEDAR) madrigal database.

The geomagnetic disturbance level for our continuum emission events was 2–4 at Kp index range, and the magnetic local time (MLT) in Svalbard is approximately UT+3h. We use the OMNIWeb solar wind and interplanetary magnetic field (IMF) measurements to assess the solar wind driving conditions for our events (Papitashvili and King, 2020).

## 3 Continuum emission observations

### 3.1 Dayside continuum on 3 January 2020

Figure 2 shows a full-colour keogram compiled of the all-sky images captured by the Sony camera at 07–11 UT on 3 January 2020. The first 1.5 hours display a combination of red emission overhead and diffuse green towards the southern horizon. The most intense pale pink continuum emissions are seen in the time frame of 08:30–09:30 UT (rectangular region in the figure), where during the first half-hour they appear particularly intense and wide. At 09:03–09:14 UT the continuum emission





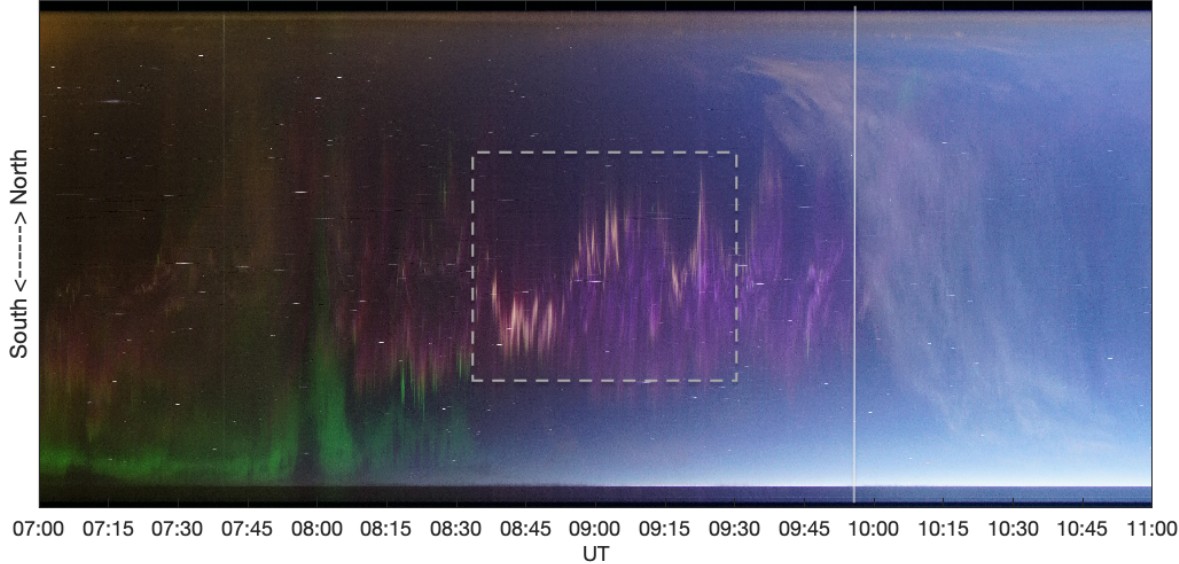

**Figure 2.** Keogram of Sony images at 07–11 UT on 3 January 2020. The y-axis is a slice through the sky from magnetic South (bottom) to magnetic North (top). The most intense pale pink continuum emission is seen in the time period of 08:30–09:30 UT indicated by the rectangle. Appearance of a thin cloud layer is marked by vertical line.

structures were very thin, but another wide continuum emission was observed at around 09:15 UT, and a continuum-bracketed corona was seen at around 09:21 UT. From about 09 UT until about 10 UT the dominant emission colour in the aurora is pink due to resonant scattering (Shiokawa et al., 2019), and the last hour is obscured by thin clouds (right of the vertical line). Some
135 individual thin arcs including the pale colour of the continuum emission took place earlier and later but the structures were too thin to be differentiated from the background in our spectral measurements.

During the time of the continuum emission observations, the IMF was moderately and steadily negative (Z component about -5 nT) as seen in Figure 3. The solar wind speed was well below $400 \, \mathrm{km \, s^{-1}}$, while the proton density was above $15 \, \mathrm{cm^{-3}}$ over
140 the time period of IMF turning negative at about 07:25 UT. Thereafter, the proton density decreased to about $10 \, \mathrm{cm^{-3}}$ during the continuum emission observations.

The left panel of Figure 1 shows an example of a pale pink arc-like structure captured with an all-sky during the first intense period of the continuum emission. The image is taken at 08:48:56 UT. No auroral structures can be seen poleward of the
145 continuum emission, and therefore this continuum emission was located at the poleward boundary of the auroral oval. This pale pink structure crossed the centre meridian of the images. The MISS spectrograph measured a clear emission enhancement across the entire wavelength region at 08:49 UT (right panel in Figure 1). We compare the continuum emission spectrum at the





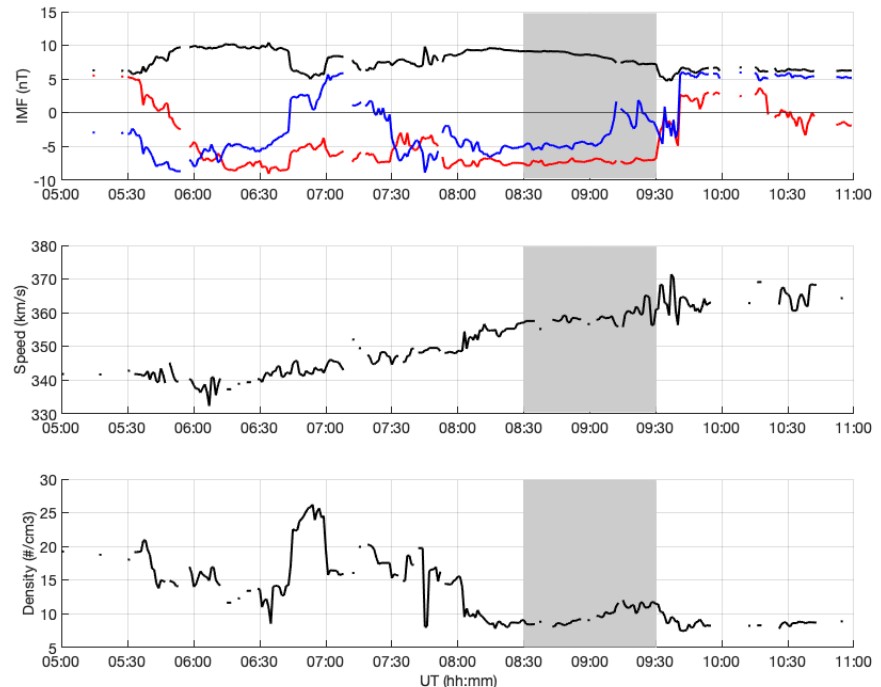

**Figure 3.** Solar wind and IMF parameters on 3 January 2020. Top: IMF magnitude (black), By (red), and Bz (blue) in nT. Middle: Solar wind speed in km/s. Bottom: Solar wind proton density in the number of particles per cm$^3$. Shaded region indicates the time period of most intense continuum emission observations. These measurements are propagated to the bow shock.

meridian location of the black square in Figure 1 to a reference spectrum that is taken from the north side of the continuum and auroral emissions (meridian location of the red square in Figure 1). The reference spectrum location is chosen so that there is no obvious emission in the sky as seen in the all-sky image. The reference spectrum is represented by the red curve in Figure 4 and only shows mild enhancements at 558 and 630 nm. The continuum spectrum (black curve in Figure 4) shows strongly enhanced auroral emission lines at 428, 558 and 630 nm in addition to elevated brightness counts throughout the measured wavelength range.

At the time the image and spectra in Figures 1 and 4 were captured, the solar elevation angle was estimated to be about -13°. This places the shadow height at about 160 km making all dayside auroral emission sunlit, which explains the purple hue of the auroral display due to resonance scattering. The shadow height estimates for the dayside events displayed in this study have been calculated by the following geometry with the assumption of a spherical Earth: $h = R_{\mathrm{E}}/\cos(\theta - 90) - R_{\mathrm{E}}$, where $h$ is the shadow height, $R_{\mathrm{E}}$ is the radius of the Earth (6371 km), and $\theta$ is the solar zenith angle in degrees.

In the continuum spectrum in Figure 4 (black curve), we can identify some specific spectral features. The blue auroral emission at around 428 nm, which is the signature of sunlit aurora, is strongly enhanced. Another milder enhancement at about




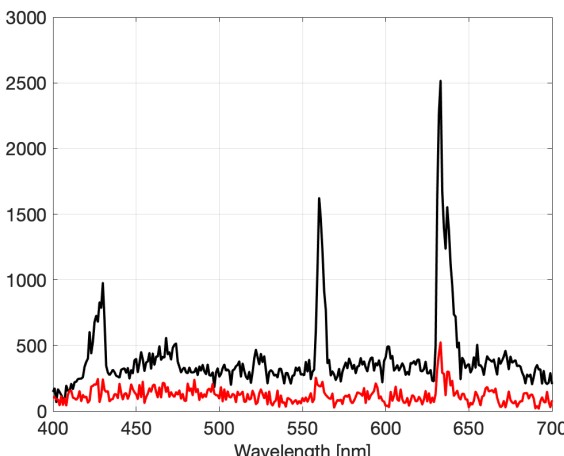

**Figure 4.** Continuum spectrum (black) and an empty-sky reference spectrum (red) measured by MISS at 08:49 UT on 3 January 2020. The y-axis values are counts summed over scan angles around the continuum maximum (118–128) and scan angles of 80–90 for the reference spectrum. The solar elevation angle was about -13° and the ionosphere above about 160 km altitude was sunlit.

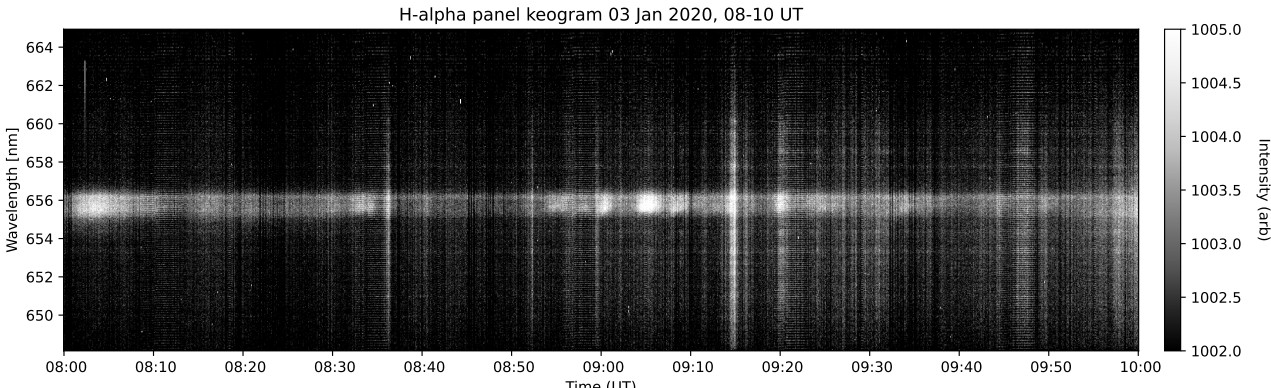

**Figure 5.** HiTIES data on 3 January 2020: the y-axis is the spatially integrated spectrograph slit, in 0.5 s second increments along the x-axis. The keogram shows proton precipitation Doppler shifted over ∼ 1nm from the rest Hα wavelength at 656.3 nm, occurring in short bursts during the event. The continuum emission crosses the spectrograph slit, and is visible as a vertical bright line at, for instance, 09:15 UT.

470 nm is also most likely from the $N_2^+$ first negative band with the nominal wavelength of 471 nm. At about 520 nm, a 10 nm wide feature may consist of both the atomic nitrogen line at 520 nm and the $N_2^+$ first negative band at 522 nm. Further in the long wavelength regime, a narrower emission signature at around 600 nm can be due to an enhancement of $O_2^+$ band at 602.6 nm (Chamberlain and Oliver, 1953; Chamberlain, 1961).

The HiTIES spectrum for the proton band (Hα) shows an increase in the proton emission during the continuum event, as seen in Figure 5. The proton precipitation is visible as a wide, bright band of proton aurora. It is blue-shifted from the Hα rest





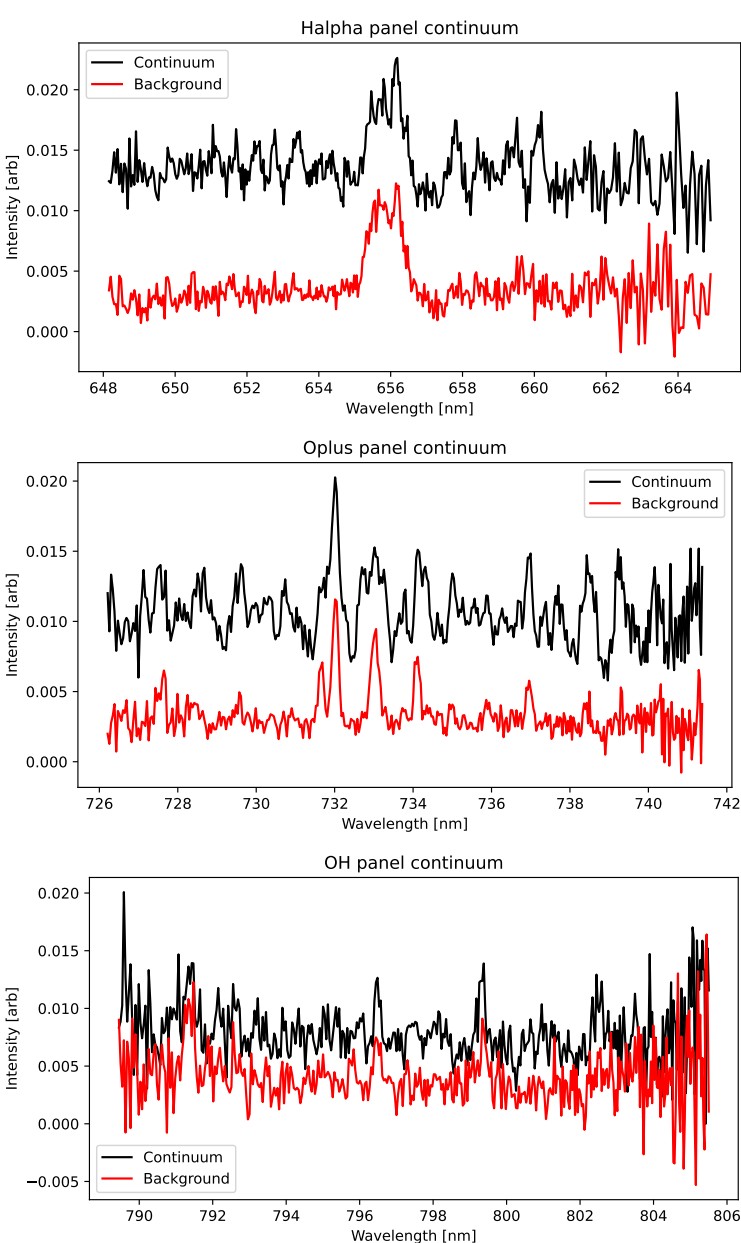

**Figure 6.** High-resolution HiTIES spectra of continuum (black) of 3 January 2020, at 09:15 UT, compared to background (red) taken from 09:11 UT same date. The measured intensity in arbitrary units is plotted against the wavelength in nm.





wavelength at 656.3 nm due to the downward velocity of the precipitating protons. The proton aurora brightens in short bursts throughout the event, including a bright burst as the continuum passes through the HiTIES slit at 09:15 UT at the centre of the ASC field-of-view (image in the top panel of Figure 8). This variation may be temporal or spatial, though it is not possible to determine which. High-resolution spectra of the continuum from HiTIES at 09:15 UT are shown in Figure 6 and compared to a reference spectrum taken at 09:11 UT, when only proton aurora emissions were within the instrument's field-of-view. These spectra show clearly the continuum nature of the emission in the observed bands, as well as some specific emission features. The background features are proton aurora (top panel), $O^+$ aurora (centre panel) and OH airglow (centre and bottom panels), these emission lines do not appear to be affected by the presence of the continuum. From the OH panel we can see that the continuum is also present in the very near-infrared. Since these data are not calibrated for absolute intensity, we can instead use the relative intensity of the continuum compared to the background to observe that the continuum becomes less bright towards the longer wavelengths, i.e. tapering off towards the infrared.

At the time of the main continuum emission observations (08:30–09:30 UT), EISCAT data shows transient enhancements of electron density which were mainly limited to the F-region (top panel in Figure 7). E-region electron density was very low, making all E-region observations highly uncertain. Particle heating can be seen as electron temperature increases at and above 150 km (2nd panel). Ion heating is also mainly concentrated in the F-region heights above about 150 km (3rd panel). Ion heating is interpreted as Joule heating, which generally requires horizontal electric field. It is not obvious if the heating is directly related to the particle precipitation and to the pale pink emission that is drifting into the EISCAT beam, or if the timing is just a coincidence. The EISCAT data were analysed with a Bayesian filtering method (BAFIM, Virtanen et al. (2021)) with an inclusion of an F-region chemical model Flipchem (Virtanen et al., 2024). This chemical model inclusion allows a variable ion composition ratio (4th panel) and therefore produces more reliable electron and ion temperatures. Ion upflow is seen to correspond the ion heating (bottom panel), mainly above 200 km.

Individual ionospheric profiles and images captured at 09:10:30 and 09:15:30 UT are shown in Figure 8. These are close to the times the HiTIES spectra was taken for the background and the continuum emissions in Figure 6. Initially (a–c), the radar beam points (green circle in panels c and f) into faint red emission region corresponding to mildly increased electron density above 200 km. Later (d–f), the beam points into the continuum emission. The electron density profile within the continuum is less enhanced in the F region than in the non-continuum profile. The same applies to the ion temperature (blue curves in b and e). Both time instances comprise soft electron precipitation. However, electron temperature is enhanced by about 500 K at about 200–250 km at the time of the continuum (e) as compared to the non-continuum profile (b).

FPI measurements of red emission can be used to estimate thermospheric neutral temperatures and winds at the heights of red emission. Figure 9 shows the comparison between the EISCAT measurements of F-region ion temperature time series from the heights of 200, 213, 229 and 246 km (top panel) and the thermospheric neutral temperature measured by FPI (bottom panel). FPI temperatures are estimated in the vertical line-of-sight direction, which is the closest direction to the field-aligned ion temperature measurements. The temporal correspondence between the two is very good. There is more variability in the ion







**Figure 7.** EISCAT Svalbard Radar (ESR) data on 3 January 2020. Electron density (top), electron temperature (2nd), ion temperature (3rd), ion composition ratio (4th) and ion drift velocity (bottom) as a function of time and height are analysed by Bayesian filtering method (BAFIM) using chemical Flipchem model.





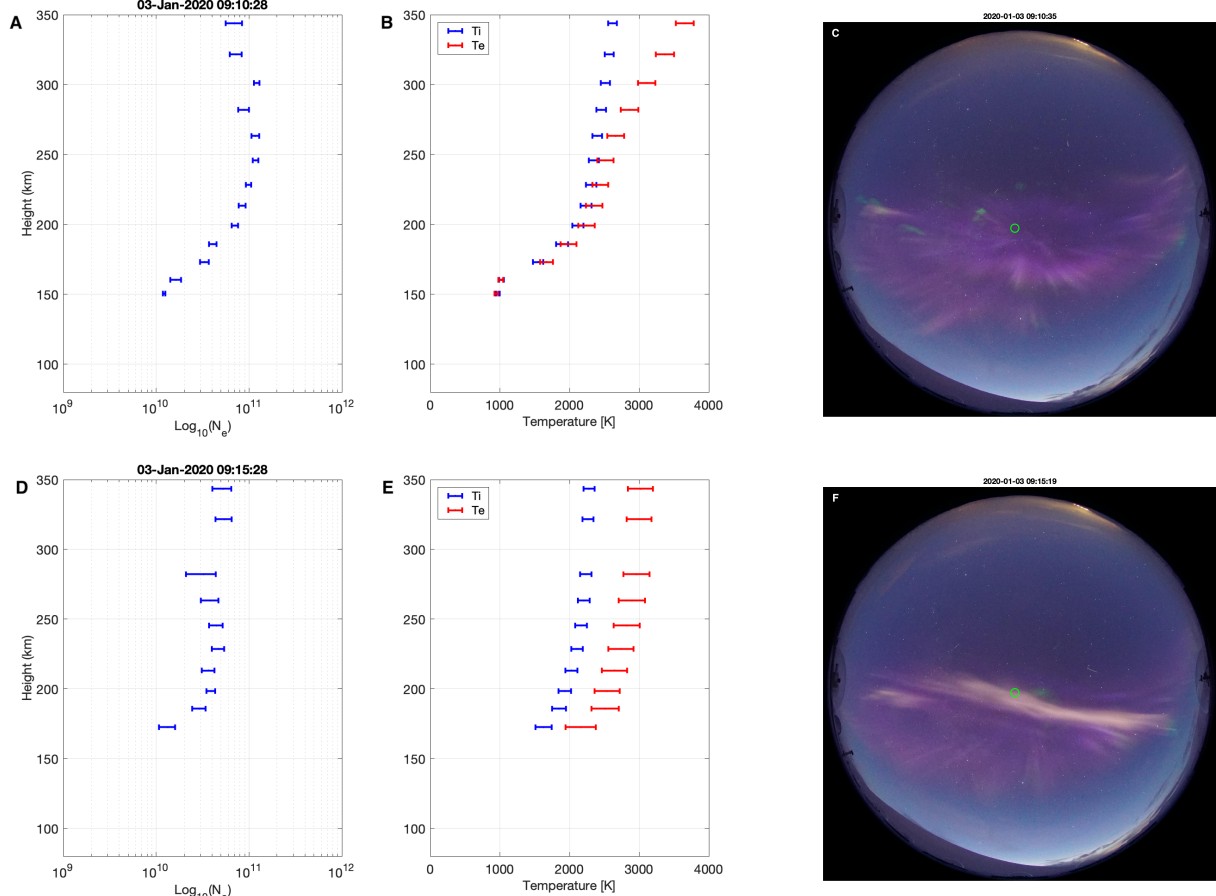

**Figure 8.** A & D: Electron density profiles measured by ESR at 80–350 km on 3 January 2020. The scale is logarithmic and all electron densities below $10^{10} \text{m}^{-3}$ have been omitted as uncertain. B & E: Ion (blue) and electron (red) temperature profiles (in K) for the same height range and the same time instants. All temperature values corresponding to heights of omitted electron densities have been excluded. C & F: All-sky images taken closest to the ESR measurements. ESR profiles times are 09:10:28 UT (A & B) and 09:15:28 UT (D & E). The corresponding image times are 09:10:35 UT (C) and 09:15:19 UT (F). At 09:15 UT the Sun was located at an elevation angle of about -12° relative to the horizon with an azimuth angle of 155° (SSE). The ionosphere was sunlit above the altitude of 145 km. The green circle in the images marks the approximate pointing direction of the radar beam at 150 km altitude.

temperature but the initial enhancement takes place at the same time (at about 07:52 UT). The initial temperature for ions and neutrals is similar (around 700–750 K), the temperatures peak closely at the same time (at about 08:56 UT) and decay during the same time period, although the neutral temperature decay takes about an hour longer (until about 10:30 UT). The maximum temperature for ions is about 2.5 times higher than the neutral maximum temperature, in agreement with Aruliah et al. (2010), who pointed out that the neutral temperature provides the lower boundary for the ion temperature variability. The FPI data also shows neutral upwelling in the vertical line-of-site direction in the time frame of ion upflow, as shown by Figure 10. The radar



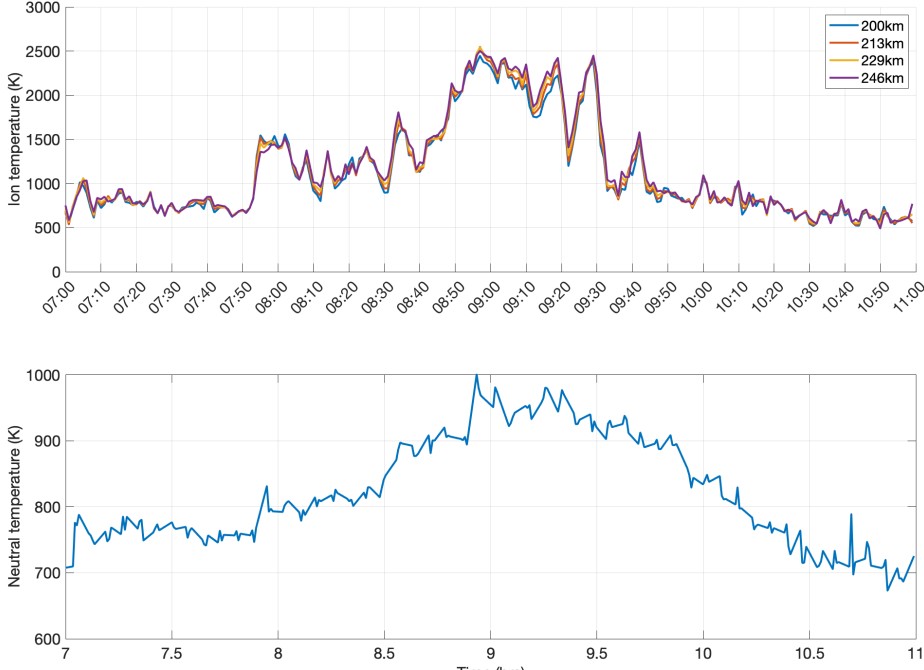

**Figure 9.** F region ion temperature measurements by ESR (top panel) at the heights of 200, 213, 229 and 246 km on 3 January 2020. Thermospheric neutral temperature measured by FPI (bottom).

measurements are displayed for four consecutive heights in the range of the strongest heating and upflow. The neutral upflow reaches higher speeds than the ions by about 40 m/s, but only for a very short period (about 10 min) just before 09:00 UT. The neutral upflow correlates best with ion upflow at 246 km height, which is a likely height for the red auroral emission to occur. The enhanced ion drift continues for about half an hour after the neutral flow had decayed. Observed upflow speeds and temperature enhancements are similar to previous upflow measurements at polar cap boundary of the auroral oval during moderate geomagnetic activity (e.g. Innis et al., 1997). Also, neutral and plasma upflow have been observed to correlate in the vicinity of auroral activity (e.g. Shinagawa et al., 2003). The flow and heating observations during the continuum emission are therefore not exceptional or extreme.

The DMSP F17 spacecraft flew over the region of continuum emission at 08:39–08:40 UT at an altitude of about 800 km. Its footpoint locations are plotted on the ASC image taken at 08:40:05 UT in Figure 11a. This image is selected as it is the closest in time to the DMSP overpass. The thin white line linearly connects the footpoint locations at 08:39 and 08:40 UT. The image is converted to geographic coordinates assuming an emission height of 150 km, which is at the bottom of the heated height range measured by EISCAT. The continuum emission moved southwest in the time between the two DMSP footpoint





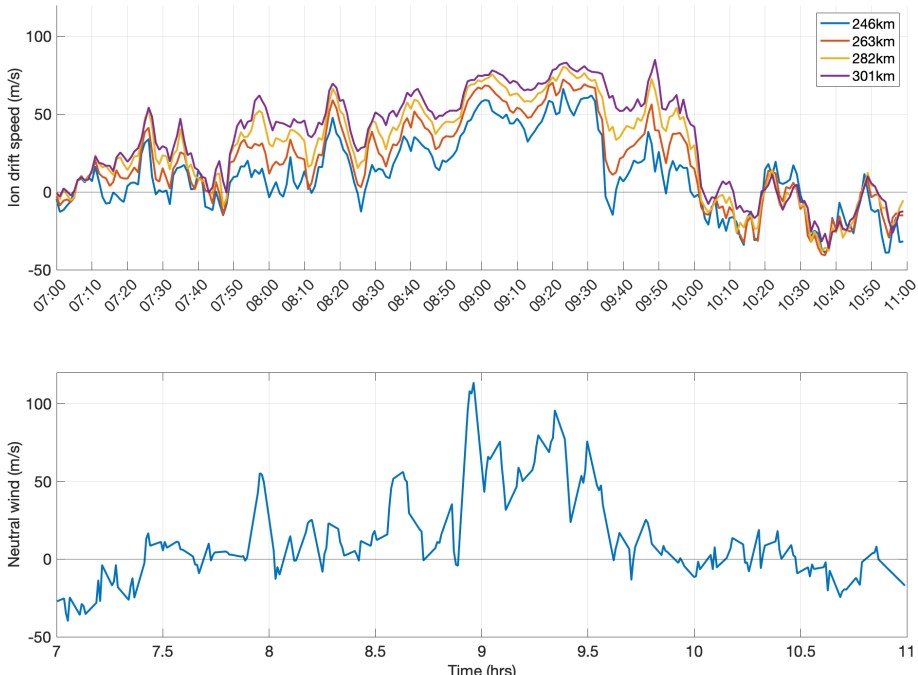

**Figure 10.** F-region ion drift speed measurements by the ESR (top panel) at the heights of 246, 263, 282 and 301 km on 3 January 2020. Thermospheric neutral upflow speed (vertical line-of-sight) measured by the FPI (bottom). Positive speed is upwards in both measurements.

locations, so the spacecraft is estimated to have crossed the continuum at 08:39:40 UT (northeastward of the location seen in the image). This time is marked by a vertical line in the plot of velocities and temperatures measured by DMSP in Figure 11b and c. Figure 11b shows the vertical (red) and horizontal cross-track velocity (blue), where the positive directions are upward

and sunward (northwest here). Figure 11c shows the ion (red) and electron (blue) temperatures. The ion temperature varies around 2000 K and the electron temperature around 4000 K, which are in agreement with the EISCAT measurement some 400 km further down (Figure 7). While the magnitude of the vertical flow measured by DMSP may not be reliable, the flow direction is upwards (positive). The horizontal cross-track flow measured by DMSP experiences a strong shear from negative (southeast) to positive (northwest) over the time of the continuum overpass. The cross-track shear flow converges towards the

region of continuum emission, which is in agreement with the upflow measured by EISCAT.

### 3.2 Morning continuum on 11 February 2024

Another continuum emission event was observed over Svalbard on 11 February 2024, but this time in the morning sector. During this event, solar wind density peaked extremely high (~40 #/cm3) right after 02:00 UT when optical instruments were





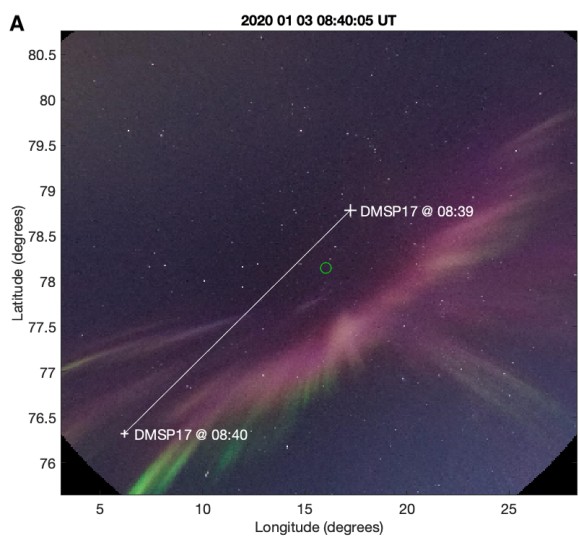

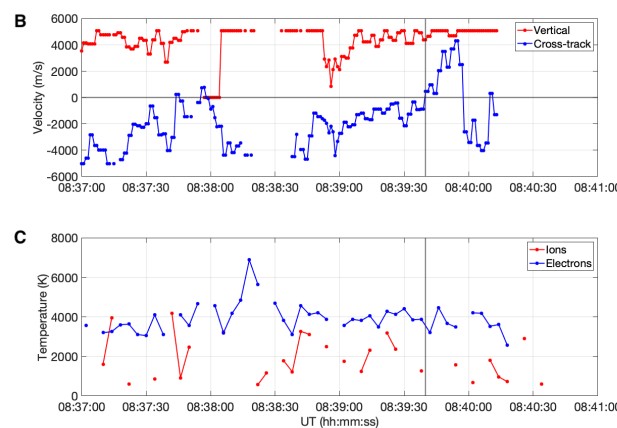

**Figure 11.** A: DMSP F17 overpass through the arc-like continuum emission. The white crosses mark the footpoint locations of the spacecraft at 08:39 and 08:40 UT. The time-wise closest ASC image at 08:40:05 UT has been plotted on geographic coordinates with the height assumption of 150 km and a linear lens model. B: DMSP measurements of vertical (red) and cross-track (blue) velocities during the overpass at 08:37–08:41 UT. Positive flow directions are upward and sunward (northwest here). C: DMSP measurements of ion (red) and electron (blue) temperatures during the overpass. The time of the estimated continuum crossing is marked by a vertical line at 08:39:40 UT.

in the dawn sector. Some continuum-like thin structures were visible as early as 02:29 UT. Strong red emission could be
observed despite variable cloudiness from about 02:40 UT onwards. The first obvious continuum signatures were measured in
the emission spectra at 05:23 UT on the southern part of the sky. These were wide arc and cloud-like structures of pale pink
emission within bright pink sunlit aurora (Figure 12). By about 06:00 UT the region of auroral emission had returned to the
southern horizon and by 06:30 UT it had become undetectable due to daylight. As long as any emission colours were visible,
the continuum emission was also present.

The MISS spectrograph data shows a particularly big difference (∼800 counts) between the continuum (black) and the
background (red) for this event. This is illustrated by the continuum emission and reference spectra at 05:46 UT in Figure 12a
and b. The sky is lighter in the region of the continuum emission than in the region of the reference spectrum due to the
increasing daylight towards the southern horizon (Figure 12c). Some of the spectral difference at the short wavelength end
of the spectrum (<500 nm) can therefore be due to the daylight, while the broad band enhancement is due to the continuum.
Most of the individual emission bands that were identified in the spectrum of the previous event are visible in this case as well,
except the enhancement of the $O_2^+$ band at 602.6 nm.

Prior to and during the continuum emission observations, the IMF Bz was negative (around -10 nT) for an extended period
of time (see Figure 13). The solar wind speed increased from about 400 km/s to about 700 km/s during the time period of the





**Figure 12.** A: MISS spectrogram data at 05:46 UT on 11 February 2024 with the meridian locations of the reference spectrum (red) and the continuum spectrum (black). B: Continuum spectrum (black) and an empty sky reference spectrum (red) measured at 05:46 UT. The y-axis values are summed counts over scan angles of 163–173 for the continuum and over scan angles of 100–110 for the reference. C: ASC image taken at 05:46:03 UT. North is to the top and east to the left of the image. The solar elevation angle was -12° relative to the horizon with an azimuth of 102° (ESE) and the sunlit boundary of the ionosphere was at 142 km.

continuum emission observations. The proton density in the solar wind peaked at values above 30–40 cm$^{-3}$ previously but
declined to values below 10 cm$^{-3}$ before the strongest continuum emissions were observed in the dawn sector.



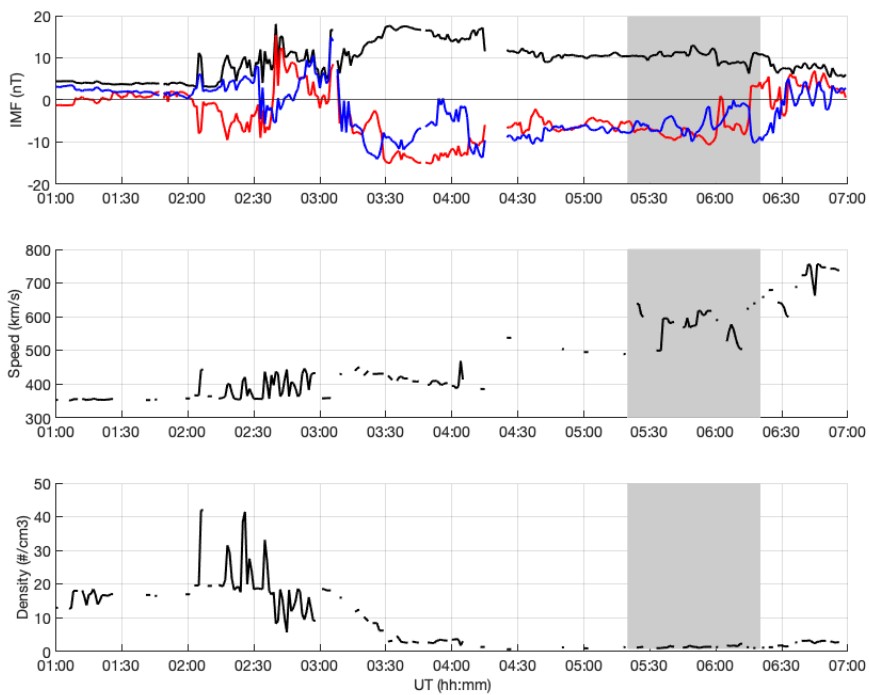

**Figure 13.** Solar wind and IMF parameters on 11 February 2024, propagated to the bow shock. *Top:* IMF magnitude (black), By (red), and Bz (blue) in nT. *Middle:* Solar wind speed in km/s. *Bottom:* Solar wind proton density in the number of particles per cm$^3$. The shaded region indicates the time period of most intense continuum emission observations.

### 3.3   Afternoon continuum on 1 December 2023

Strong red auroral emission was observed on 1 December 2023, particularly in the afternoon. Most of the continuum emission structures were either off the meridian / zenith or were very thin structures. The spectral comparison between the continuum and the background sky, such that was done for the previous events, was therefore not possible. The colour of the observed emission, however, strongly suggests the structure was a continuum emission. The thin continuum-like structures appeared as early as about 08:30–10:00 UT, while more clearly field-aligned structures were seen at about 15:00–16:00 UT. Some of the continuum-like rays were captured by photographers. The photograph in Figure 14b was taken independently by a citizen scientist and captured the same thin structure observed in Figure 14a. This photo provides a side view into the thin continuum emission clearly showing the field-aligned orientation of the thin pale structure. The all-sky images show the thin continuum emission persisting for a minute, from 15:11:10 to 15:12:10 UT. Similarly the pale pink rays seen in Figure 14c were clearly visible in the ASC images for about a minute.




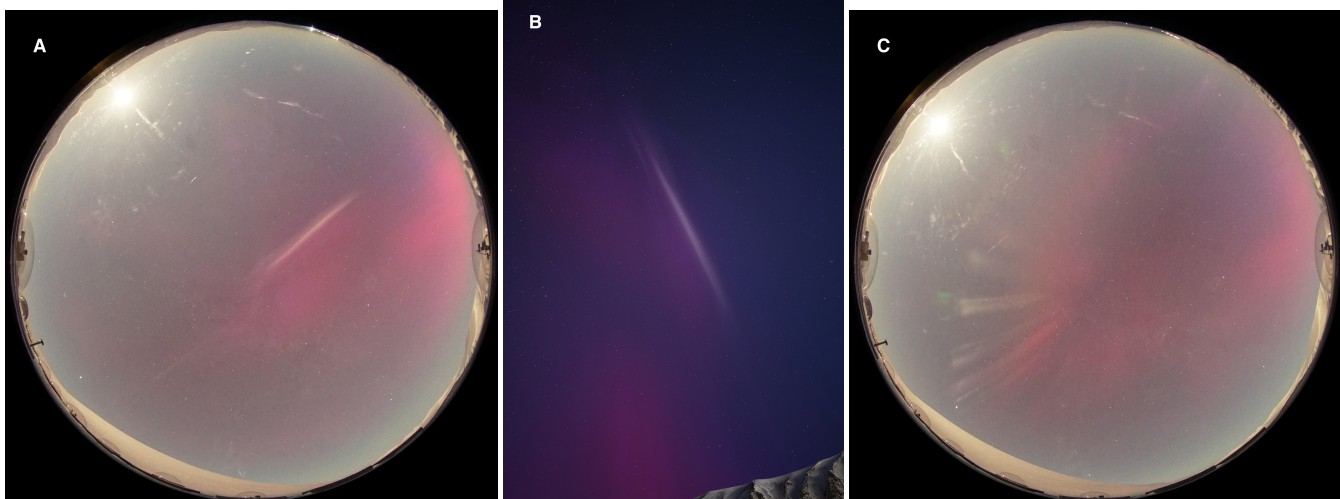

**Figure 14.** Example Sony images of the afternoon continuum taken on 1 December in 2023 at 15:11:58 UT (A) and 16:00:56 UT (C). The thin arc in panel A was shown to be a field-aligned structure based on the photograph taken independently in the same region and on the same minute by Marjan Spijkers (B). The photo was taken with 8-second exposure time and viewing the sky towards west. Pale bundles of field-aligned rays are seen in the image C. North is to the top and east to the left in ASC images (left and right). At 15:11:58 UT the Sun was located at an elevation angle of -17° with an azimuth of 243° (WSW). The ionosphere was sunlit above an altitude of 284 km.

The infrared camera NIRAC also detected the thin emission structure, which overlaps with the continuum-like field-aligned structure in the left panel of Figure 14. The NIRAC image displayed in Figure 15 shows no enhancements related to the surrounding red emission (630 nm) but an enhanced thin structure of the continuum at 1.1 $\mu$m. This demonstrates that the wide spectral enhancement of the continuum emission may not be limited to the visible wavelength range but may extend deep into the infrared regime. During this event, NIRAC was operated at 5-second cadence. This infrared emission from molecular nitrogen is likely to be caused by charge exchange from atomic oxygen ion as it is likely occurring in the F-region together with the surrounding red emission.

Prior to and during the continuum emission observations the IMF magnitude was relatively stable and high (around 20 nT) as seen in Figure 16. The Z-component of the IMF was strongly negative (about -20 nT) until about 13:20 UT. The solar wind speed was above 500 km/s throughout most of the time prior to and during the continuum emission. The proton density in the solar wind varied primarily above 20 $\mathrm{cm}^{-3}$, reaching the peak values above 50 $\mathrm{cm}^{-3}$ between around 11:00–13:00 UT.

### 3.4 Nighttime continuum on 8 February 2019

Continuum-looking arc-like features were observed in the ASC images close to the zenith/centre meridian several times during the first hour of the day on 8 February 2019. As seen in the example images in Figure 17, these events are faint and narrow

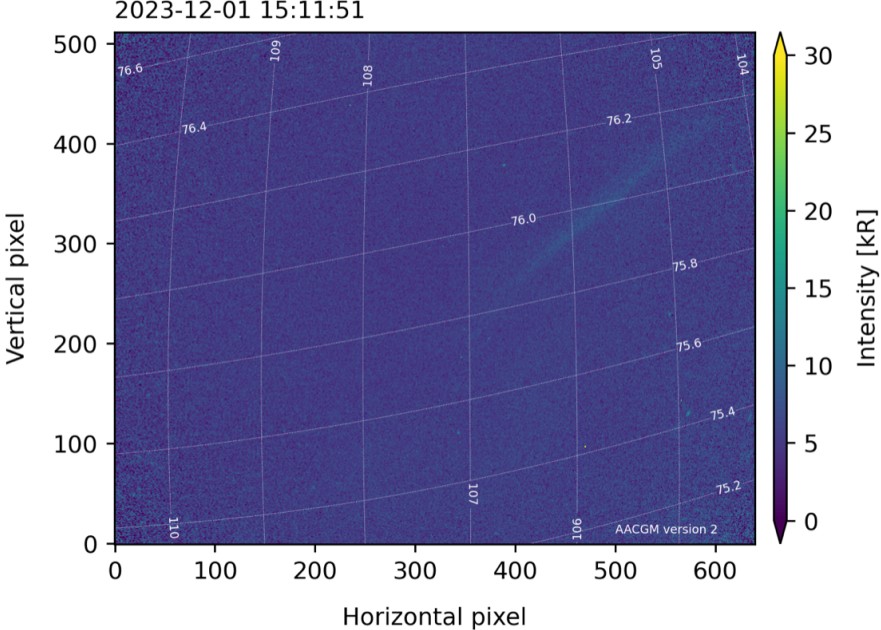

**Figure 15.** An infrared image of the thin structure in the images of Figure 14. No red emission at 630 nm is seen but the continuum structure is clearly visible at 1.1 $\mu$m. The white grid indicates the geomagnetic coordinates.

structures and therefore hard to detect even for a trained human observer. The meridian imaging spectrograph was not able
to resolve the continuum from the background sky spectrum, but the colour of the emission cannot be explained by any other
known emission than the continuum. The thin continuum-like structures developed in the image data for about 8 minutes (from
about 00:26 until 00:34 UT). The solar wind prior to and during the event was slow (about 400 km/s) with a moderate density
of 4–5 cm$^{-3}$. The IMF Z-component varied between -4 and +2 nT (data not shown).

## 4   Discussion

In this study, we present the first continuum emission observations captured near the poleward boundary of the dayside auroral
oval. We described two spectrally confirmed, intense continuum emission events on 3 January 2020 and on 11 February 2024,
in the noon and dawn sector respectively. Additionally, we show nighttime and afternoon events, not confirmed by spectral
measurements, but are classified as continuum emission based on their pale pink colour showing visual similarities to the
spectrally confirmed events. The afternoon sector event mainly manifested itself as thin field-aligned structures of continuum-
like emission. An example of a nighttime continuum-like emission event appeared as a narrow arc at the poleward boundary of
the green aurora.



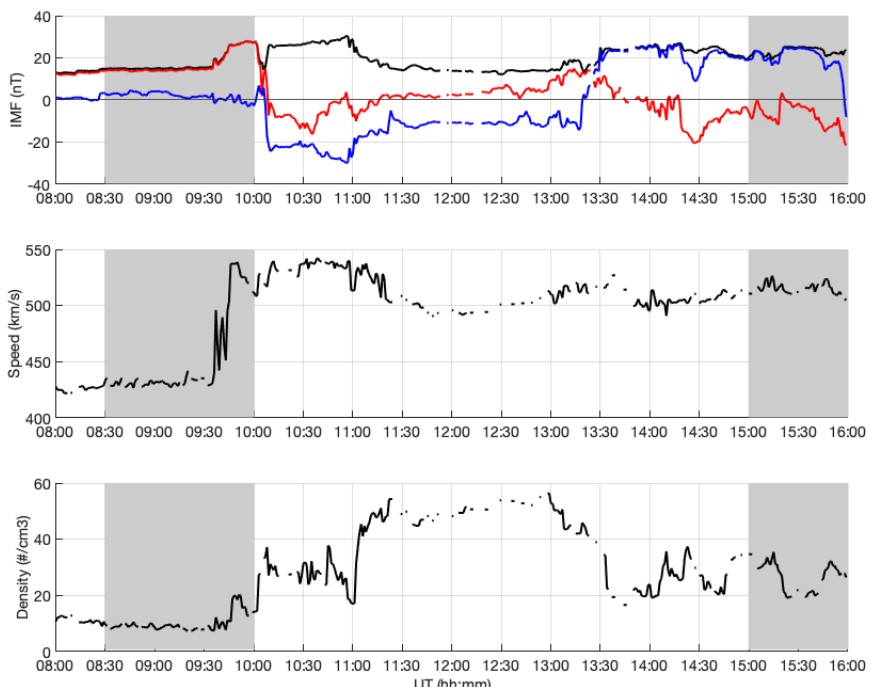

**Figure 16.** Solar wind and IMF parameters on 1 December 2023, propagated to the bow shock. *Top:* IMF magnitude (black), By (red), and Bz (blue) in nT. *Middle:* Solar wind speed in km/s. *Bottom:* Solar wind proton density in the number of particles per cm$^3$. The shaded regions indicate the time periods of most intense continuum emission observations.

Auroral conditions similar to those seen on 1 December 2023 with strong red-dominated emission and thin continuum-like emission structures were visually found at least on 12, 21, 23–27 and 30 November 2022; 1, 6 and 7 November 2023; 2–3, 6 and 15–16 December 2023, as well as 6 February 2024. These are all likely to be continuum emissions, in which case, this

type of emission is not a rarity at high latitudes. It rather seems infrequent to see it so strong and wide that it is unambiguously visible in the spectrograph data and keograms. The observed pale pink emission moves with the auroral particle precipitation and often appears as field-aligned rays that may evolve quickly in time. Because the individual continuum emission structures last typically of the order of a minute or less, the mismatch in the timing between the ASC images and the spectra plays a role in the number of spectrally confirmed observations.

Interestingly, during each of the continuum emission events shown in this study, Fragmented Aurora-like Emissions (FAEs) occurred nearby (such as in top panel of Figure 8). Fragments were first described by Dreyer et al. (2021) as localised emission regions, which lack the field-aligned extent. While there is no commonly established generation mechanism for fragments, the authors concluded that fragments were not caused directly by particle precipitation but that local instabilities would play a key role in producing the observed emission. Not all fragment observations align with observations of continuum emissions,



**Figure 17.** Example images of our only nightside continuum event on 8 February 2019. This continuum emission lasted for about 10 minutes and showed a temporal evolution like any auroral event. Images are taken at 00:26:55 (A), 00:27:55 (B) and 00:28:43 UT (C). North is to the top and east to the left in the images. The Sun's elevation angle was -26° relative to the horizon and the shadow height was above 700 km.



but vice versa seems to be true, suggesting that the strong ionospheric heating may lead to favourable conditions for fragment formation.

The behaviour of the continuum emission of STEVE is different from our observations of the high-latitude continuum emission. STEVE typically appears before the magnetic midnight, relates to the substorm recovery phase, and lasts for about an

hour (Gallardo-Lacourt et al., 2018a). In contrast, our examples of continuum emissions are transient structures with lifetimes of minutes, distributed across different magnetic local times, from midnight to dayside cusp hours and late afternoon. Apart from the midnight event, our continuum emissions take place in time sectors multiple hours away from substorm activity. While the STEVE continuum is often seen as a narrow horizontal arc, our events show variable morphological structures from thin to wide arcs, perturbed arcs and field-aligned rayed structures. However, field-aligned rays are sometimes observed as a

fine-structure of STEVE too, so the rayed continuum emission structure is not fully absent, just perhaps not as frequent in STEVE as it is among the events reported in this study. The dynamic behaviour of the continuum emission events reported here is similar to the continuum emission events found embedded in the auroral oval (Spanswick et al., 2024). Also similar patchiness is seen in our continuum emission events as reported by Nanjo et al. (2024). One of the structures reported here was related to a strong flow shear measured at low-Earth orbit, comparable to the extreme sub-auroral ion drift associated with

STEVE (MacDonald et al., 2018; Archer et al., 2019a). It remains to be discovered if strong flow shears are also commonly associated with high-latitude continuum emissions.

According to Nishiyama et al. (2024), the molecular nitrogen Meinel band in the infrared range can be excited by charge exchange with atomic oxygen ions. In particular, the excited D state of the atomic oxygen may be an important source of ionised molecular nitrogen emission in the infrared range ($N_2^+$ Meinel band). If this IR band was an extension of the continuum, the

measurements of this emission would support atomic oxygen ions having an important role in the continuum process, whether they were produced by particle precipitation or enhanced through ion upflow. Furthermore, spectral measurements between the proton bands and the $N_2$ Meinel band deeper in the infrared would be helpful to confirm if the continuum emission truly extends all the way to over 1100 nm, or if the continuum emission and the enhanced IR Meinel band measurements coexist due to another reason during this particular event. As indicated in the discussion of Figure 4, the observed broad-band spectral

enhancement may build up from contributions of many thermally excited species. Further work on spectral measurements in combination with emission modelling is required to fully reveal the contributions of different emitting species. Also careful examination of the heating and energy transition processes are needed to fully resolve which conditions are required for the continuum emission to appear and how wide of a wavelength region it can involve.

During the continuum event on 3 January 2020, the ESR measured the strongest ion heating above about 200 km and

electron heating above about 250 km. Ion temperatures reached about 2500 K at 150–350 km heights during our continuum event, which is similar to the values that have been measured during the STEVE continuum (Archer et al., 2019b; Liang et al., 2019). The heating-induced emission that was proposed to explain STEVE emission is based on availability of vibrationally excited molecular nitrogen that produces NO, which further reacts with atomic oxygen to result in $NO_2$ and light (Harding et al., 2020). The brightness of the resulting $NO_2$ continuum was deemed sensitive to neutral heating and upwelling. In our case



the upward ion drift is strongest above 150 km. The electron precipitation during the observed continuum emission occasionally reaches 150 km. This places both enhanced electron density and the upward moving ions in the height range typical for the red-dominated auroral emission. More detailed observations and modelling of this region is required to judge if the observed heating and upwelling can produce enough $NO_2$ continuum to explain the observed emission.

During the afternoon of 1 December 2023, when thin field-aligned continuum emissions were observed, the ESR measured

particle precipitation and heating of the same order of magnitude as those seen during the continuum on 3 January 2020 in Figure 7 (electron temperatures up to about 3000–4000 K and ion temperatures up to about 2000 K at 300 km). However, no continuum emission was observed within the radar beam or in the vicinity. This may either suggest that an additional local heating mechanism is needed to produce the continuum, or that particle and Joule heating are required to reach down to at least 200 km altitude before sufficient atmospheric composition is available for the production of the continuum emission.

A common driver for the two strong dayside continuum emission events as well as the afternoon continuum described in this study may be the high solar wind particle density during or before the event of continuum emissions. Kataoka et al. (2024) studied the storm on 1 December 2023, which had a moderate Dst but very intense red auroral emission seen at exceptionally low latitudes. They concluded that high solar wind density (pressure) drives the strongly red-dominated auroral emission by compression of the magnetopause. This intense red emission was observed in our continuum events as well. For our nighttime

continuum event, the driver is less obvious. This event is also much less prominent than the dayside events. While large numbers of particles from the solar wind entering the magnetosphere–ionosphere system could indeed heat the upper atmosphere, the ionospheric and solar wind driving conditions for the events presented in this study are very different. Therefore, further research is required to resolve the necessary conditions to facilitate continuum emission.

## 5   Conclusions

We present the first observations of high-latitude dayside continuum emissions. The events, which were confirmed to be continuum emissions by spectral measurements, were seen at the polar cap boundary of the dayside auroral oval. We additionally presented visually identified continuum-like events across other MLT sectors. These events were classified as continuum based on the observed emission colour, which cannot be explained by any other known emission. All continuum (and alike) emission events shown in this study involve a range of different ionospheric conditions. For the one event for which ESR, FPI and DMSP

data are available, we see enhanced temperatures in both electron, ion and neutral particle populations. These temperature increases suggest both precipitating particles and Joule heating to be present. We also identified a strong horizontal plasma flow shear in the region of the continuum emission, which may further contribute to the heating and upflow of ions and neutrals.

Our measurements reveal that a number of different emission lines and bands contribute to the broadband continuum emissions. Even emission structures in the near and far infrared were observed, which suggest multiple emitting species to be

responsible for the observed spectral enhancements.



Resolving the exact spectral structure and chemical constituents of the continuum will require further investigations, involving both observations and modelling. Also, understanding the required ionospheric conditions for continuum emission will be examined in the future with more observational evidence.

*Code availability.* Software for wavelength calibration and plotting of MISS data is available at https://github.com/UNISvalbard/KHO-
MISS/ . The star calibration and mapping of Sony images is done with software at https://github.com/UNISvalbard/KHO-starcalibration .
The BAFIM software (Virtanen et al., 2024) is available from https://doi.org/10.5281/zenodo.4033903. The Flipchem Python interface to the
IDC package (Reimer et al., 2021) is available from https://doi.org/10.5281/zenodo.3688853.

*Data availability.* Solar wind data were downloaded from OMNIweb (https://omniweb.gsfc.nasa.gov). DMSP data is available through
CEDAR Madrigal database http://cedar.openmadrigal.org . EISCAT data can be accessed through https://portal.eiscat.se/schedule/ . MISS,
Sony, HiTIES, and FPI datasets of this study are available as doi: 10.5281/zenodo.13960606 .

*Video supplement.* Animation of Sony image on 3 January 2020 at 7–11 UT is included in the article dataset, doi: 10.5281/zenodo.13960606
.

*Author contributions.* NP, RDO and KH conceptualised the study, selected the events and wrote most of the manuscript; BGL wrote the
introduction and high-latitude continuum comparison with STEVE observations; TN provided DMSP data and helped with its interpretation;
RDO and DW analysed proton precipitation data and helped with the interpretation of the continuum spectra; IV analysed ESR data for
this study and helped with its interpretation; TN provided IR spectra for the continuum and helped with the interpretation of the continuum
spectrum; MB contributed to the interpretation and modelling of the continuum spectra; FS helped with the technical interpretation of MISS
data and run the instrument calibration for this work; MiS provided the MISS data access, helped with cleaning, plotting and wavelength
calibration of those data; AA provided the FPI data and assisted the analysis; MaS provided and processed the photo of a FA continuum; LM
calculated the solar elevation angle and shadow heights. All authors have participated in the discussions of the observations, the interpretation
of the results and editing of the manuscript.

*Competing interests.* The authors report no competing interests.

*Acknowledgements.* This research was supported by the International Space Science Institute (ISSI) in Bern, through ISSI Working Group
project ARCTICS. The authors thank Sony camera PI Dag Lorentzen for the use of images. KH and LM are financed by the Norwegian
Research Council under contract #343302. The work of DW is supported by the Natural Environment Research Council of the UK (grant



NE/S015167/1). MG is supported by the Research Council of Finland grants 338629-AERGELC'H and 360433-ANAON. We acknowledge the EISCAT Scientific Association for the radar data. EISCAT is an international association supported by research organisations in China (CRIRP), Finland (SA), Japan (NIPR and ISEE), Norway (NFR), Sweden (VR), and the United Kingdom (UKRI).



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
