# Peer review of "First observations of continuum emission in dayside aurora"

_EGUsphere, 2024_

## Author Response (AR2)

**RC1: 'Comment on egusphere-2024-3669', Anonymous Referee #1, 07 Jan 2025**

*This manuscript reports on a phenomenon referred to as "continuum emission," which appears as pink or purple light. While the color is reminiscent of the known phenomenon STEVE, the differences in its occurrence location and morphological characteristics ensure its novelty. The paper introduces observational results from various instruments, including a commercial digital camera, spectrometer, incoherent scatter radar, Fabry-Perot interferometer, the DMSP satellite, and OH imager, all of which are explained in detail. While the mechanism underlying all presented cases requires further clarification through future observations and modeling, the authors discuss various plausible explanations. For example, a flow shear in plasma shown in Figure 11, which is also observed in STEVE and its dawnside counterpart, is a valuable discovery that brings us closer to understanding the generation mechanism of continuum emissions. I do not have significant disagreements with the content of the manuscript. However, I believe the following revisions would make the paper more accessible and comprehensible to readers:*

*1. When comparing the spectra of continuum and the background in Figure 4, how about including a comparison with the spectrum of sunlit aurora as well? While Figures 1A and 12C clearly illustrate the visual color differences between continuum and sunlit aurora, a more precise comparison using spectra could help to understand their differences.*

This is a good idea that has been discussed. We want to add a spectrum that is taken from the same image/

spectrogram as our continuum and background sky spectra to avoid any changes in the sky or auroral conditions, and this seemed most feasible to do for the event in Figure 12, because the sunlit aurora structures are narrower and fainter in the event in Figure 4. We have therefore added a spectrum to Figure 12B, and its corresponding location in Figure 12A, just north of the continuum, which shows much less background and equally strong red emission as compared to the continuum, while the blue emission is seen more enhanced than the green one as compared to the background spectrum. The enhanced blue component in the dayside red aurora is signature of the sunlit conditions.

*2. To make the comparison between Figures 1A and 1B clearer, could you add numerical labels corresponding to the scan angles in Figure 1B onto the slit region (white rectangle) in the all-sky image (Figure 1A)?*

Approximate spectrograph pixel values of 40, 80, 120 and 160 have been added into Figure 1A for reference.

*3. Although the direction is explained in the caption of Figure 1A, it would be helpful to indicate "N," "S," "E," and "W" directly in Figure 1A to make it visually clearer.*

Small arrows indicating the direction to north and east has been added onto Figure 1A to support the caption text.

*4. I understand that Figure 1B is not calibrated for sensitivity, but adding a color bar might still be helpful for readers to understand the scale.*

Figure 1B is showing uncalibrated intensity, so it is just counts, just like the values on Y axis of Figure 4 and 12B. For Figure 1B, this has been clarified in the figure caption,

while for Figures 4 and 12B the axis label has been added to support the figure caption texts.

*5. Please add a label to the vertical axis of Figure 4.*

Sure, axis label has been added.

*6. The time labels in Figures 8C and 8F are too small to read. Could you enlarge them?*

Absolutely. Figure 8C and 8F time and panel labels have been enlarged significantly.

*7. Please add a label to the vertical axis of Figure 12B.*

Axis label has been added.

**RC2: 'Comment on egusphere-2024-3669', M.J. Kosch, 24 Feb 2025**

*This is a well-written and comprehensive paper describing a new type of aurora, namely, continuum emission poleward of the dayside auroral oval. The paper is novel and therefore worthy of publication. I recommend publication after a few minor revisions to improve clarity:*

*Figures 9 and 10: Please add error bars to both panels.*

Yes, error bars are included in the revised version of the manuscript, where then, for clarity, we only include two different heights. These are the top and bottom heights from the same range as before, to show the small magnitude of the vertical changes.

*Figure 15: The phenomenon described is simply not visible in the figure. If it is very faint, I suggest adding arrows to aid*

*the reader.*

Good point. This is done.

*Figure 17: A non-expert reader may easily become confused with the normal aurora present. Please add an arrow to point out the continuum event.*

Indeed. The region of interest has been marked in the revised version of the manuscript.